# Antidiabetic Therapy in the Treatment of Nonalcoholic Steatohepatitis

**DOI:** 10.3390/ijms21061907

**Published:** 2020-03-11

**Authors:** Yoshio Sumida, Masashi Yoneda, Katsutoshi Tokushige, Miwa Kawanaka, Hideki Fujii, Masato Yoneda, Kento Imajo, Hirokazu Takahashi, Yuichiro Eguchi, Masafumi Ono, Yuichi Nozaki, Hideyuki Hyogo, Masahiro Koseki, Yuichi Yoshida, Takumi Kawaguchi, Yoshihiro Kamada, Takeshi Okanoue, Atsushi Nakajima

**Affiliations:** 1Division of Hepatology and Pancreatology, Department of Internal Medicine, Aichi Medical University, Nagakute, Aichi 480-1195, Japan; yoneda@aichi-med-u.ac.jp; 2Department of Internal Medicine, Institute of Gastroenterology, Tokyo Women’s Medical University, Tokyo 162-8666, Japan; tokushige.ige@twmu.ac.jp; 3Department of General Internal Medicine2, Kawasaki Medical School, Okayama 700-8505, Japan; m.kawanaka@med.kawasaki-m.ac.jp; 4Department of Hepatology, Graduate School of Medicine, Osaka City University, Osaka 558-8585, Japan; rolahideki@med.osaka-cu.ac.jp; 5Department of Gastroenterology and Hepatology, Yokohama City University Graduate School of Medicine, Yokohama 236-0004, Japan; yoneda-ycu@umin.ac.jp (M.Y.); kento318@yokohama-cu.ac.jp (K.I.); nakajima-tky@umin.ac.jp (A.N.); 6Department of Metabolism and Endocrinology, Faculty of Medicine, Saga University, Saga 840-8502, Japan; takahas2@cc.saga-u.ac.jp; 7Liver Center, Saga University Hospital, Saga 840-8502, Japan; eguchiyu@cc.saga-u.ac.jp; 8Division of Gastroenterology and Hepatology, Department of Internal Medicine, Tokyo Women’s Medical University Medical Center East, Tokyo 116-8567, Japan; ono.masafumi@twmu.ac.jp; 9Department of Gastroenterology, National Center for Global Health and Medicine, Tokyo 162-8655, Japan; ynozaki@hosp.ncgm.go.jp; 10Department of Gastroenterology, JA Hiroshima General Hospital, Hiroshima 738-8503, Japan; hidehyogo@ae.auone-net.jp; 11Division of Cardiovascular Medicine, Department of Medicine, Osaka University Graduate School of Medicine, Suita Osaka 565-0871, Japan; koseki@cardiology.med.osaka-u.ac.jp; 12Department of Gastroenterology and Hepatology, Suita Municipal Hospital, Osaka 564-8567, Japan; 13Division of Gastroenterology, Department of Medicine, Kurume University School of Medicine, Kurume 830-0011, Japan; takumi@med.kurume-u.ac.jp; 14Department of Molecular Biochemistry & Clinical Investigation, Osaka University Graduate School of Medicine, Suita, Osaka 565-0871, Japan; ykamada@gh.med.osaka-u.ac.jp; 15Hepatology Center, Saiseikai Suita Hospital, Osaka 564-0013, Japan; okanoue@suita.saiseikai.or.jp

**Keywords:** dipeptidyl peptidase-4, fibroblast growth factor, gastrointestinal peptide, glucagon-like peptide 1, glucagon receptor, peroxisome proliferator-activated receptor, sodium glucose cotransporter

## Abstract

Liver-related diseases are the third-leading causes (9.3%) of mortality in type 2 diabetes (T2D) in Japan. T2D is closely associated with nonalcoholic fatty liver disease (NAFLD), which is the most prevalent chronic liver disease worldwide. Nonalcoholic steatohepatitis (NASH), a severe form of NAFLD, can lead to hepatocellular carcinoma (HCC) and hepatic failure. No pharmacotherapies are established for NASH patients with T2D. Though vitamin E is established as a first-line agent for NASH without T2D, its efficacy for NASH with T2D recently failed to be proven. The effects of pioglitazone on NASH histology with T2D have extensively been established, but several concerns exist, such as body weight gain, fluid retention, cancer incidence, and bone fracture. Glucagon-like peptide 1 (GLP-1) receptor agonists and sodium-glucose cotransporter 2 (SGLT2) inhibitors are expected to ameliorate NASH and NAFLD (LEAN study, LEAD trial, and E-LIFT study). Among a variety of SGLT2 inhibitors, dapagliflozin has already entered the phase 3 trial (DEAN study). A key clinical need is to determine the kinds of antidiabetic drugs that are the most appropriate for the treatment of NASH to prevent the progression of hepatic fibrosis, resulting in HCC or liver-related mortality without increasing the risk of cardiovascular or renal events. Combination therapies, such as glucagon receptor agonist/GLP-1 or gastrointestinal peptide/GLP-1, are under development. This review focused on antidiabetic agents and future perspectives on the view of the treatment of NAFLD with T2D.

## 1. Introduction

One-fourth of the adult population is now estimated to be suffering from nonalcoholic fatty liver disease (NAFLD) worldwide [1,2]. Nonalcoholic steatohepatitis (NASH), a more severe form of NAFLD, is defined by liver fat deposition with inflammation and ballooning. The incidence of NASH has risen dramatically over the last two decades because the prevalence of obesity, metabolic syndrome, and type 2 diabetes (T2D) is growing. NASH can at least partly progress to severe fibrosis, and cirrhosis over time, with a high risk for liver failure and hepatocellular carcinoma (HCC). In the US, NASH has become the leading cause of end-stage liver disease or liver transplantation [3]. In Japan, the liver-related disease is the third leading cause of mortality (9.3%) in T2D, according to a nationwide survey (2001–2010) [4]. T2D patients are at higher risk for the development of or mortality from HCC [5,6]. Therefore, NASH can be called “Diabetic Liver Disease”. It is estimated that the prevalence of diagnosed NASH will reach 45 billion US dollars by 2027 in the US, Japan, England, and the so-called EU 4 (France, Germany, Italy, and Spain). Lifestyle interventions, such as dietary caloric restriction and exercise, are currently the cornerstone of therapy for NASH, but such changes can be difficult to achieve and maintain, underscoring the dire need for pharmacotherapy. The first-line therapy for those without diabetes is vitamin E on the basis of accumulating evidence because vitamin E has prevented progression to liver decompensation or transplantation in NASH patients with advanced fibrosis [7]. In T2D patients with NASH, however, vitamin E alone does not significantly change the primary histological outcome (a 2-point reduction in NAFLD activity score (NAS) from two different parameters, without worsening of fibrosis). The combination therapy of vitamin E and pioglitazone is better than a placebo in improving liver histology in NASH patients with T2D [8]. Metformin is now the first-line pharmacotherapy for T2D according to western guidelines [9], but it has no effect on NAFLD or NASH [10,11,12,13]. Thus, there are no established pharmacotherapies for NASH with T2D, except pioglitazone [14]. The leading cause of mortality in NAFLD is cardiovascular events. NASH drugs should provide cardioprotective effects, as well as hepatoprotective effects. This review provided an overview of the role of current and novel antidiabetic agents in the treatment of NASH (Table 1).

## 2. Association of T2D with NASH and NAFLD

The pooled prevalence of NAFLD in T2D patients, obtained by a random-effects model, is 59.67% (95% confidence interval (CI): 54.31–64.92%) [17]. Based on 49,419 individuals with T2D among 80 studies from 20 countries, the global prevalence of NAFLD among patients with T2D is 55.5% (95% CI 47.3–63.7%). Among 10 studies that estimated the prevalence of NASH, the global prevalence of NASH among individuals with T2D is 37.3% (95% CI 24.7–50.0%). Seven studies estimated the prevalence of advanced fibrosis in T2D patients with biopsy-proven NAFLD to be 17% (95% CI 7.2–34.8%) [18]. Among 18.2 million people in the US living with T2D and NAFLD, 6.4 million have NASH [19]. In T2D patients, the prevalence of advanced fibrosis is estimated to be 7.3–24.9% by FibroScan and 4.3–7.1% by magnetic resonance elastography (MRE) [20]. Older age (odds ratio (OR) = 1.099, *p* = 0.001), high body mass index (BMI) (OR = 1.088, *p* = 0.003), low platelet level (OR = 0.996, *p* = 0.014), and smoking (OR = 1.653, *p* = 0.013) are independent risk factors of advanced fibrosis (FibroScan > 10.6 kPa) among T2D patients [21]. The existence of T2D is closely associated not only with advanced fibrosis in cross-sectional data [22,23,24] but also with the rapid progression of hepatic fibrosis based on longitudinal data [14,18,25,26,27,28]. Conversely, NAFLD patients have a higher risk of incidental T2D compared to non-NAFLD patients [27,28]. The annual incident rate of overt diabetes (glycated hemoglobin (HbA1c) ≥ 6.5%) is around 2% in NAFLD without T2D when the 75 g oral glucose tolerance test is used to confirm the absence of diabetes at entry [26]. Insulin resistance in NAFLD leads to incident T2D [24,26]. In conclusion, T2D and NAFLD are mutually, closely, and bi-directionally associated [25].

## 3. Peroxisome Proliferator-Activated Receptors

Peroxisome proliferator-activated receptors (PPARs) are nuclear receptors that regulate lipid and insulin metabolism. Pioglitazone (PPAR γ agonist) shows a statistically significant improvement in NASH compared to with placebo [29,30,31]. However, pioglitazone has several concerns for practical clinical use, such as the increased risk of body weight gain, fluid retention, increased cardiovascular events, prostate cancer, pancreas cancer, and bone fracture, in post-menopausal women. INT131 is a selective PPARγ modulator under development for T2DM patients. Dose-dependent reductions have been observed in HbA1c, equivalent to 45 mg pioglitazone, but with less fluid accumulation and body weight gain [32]. No study with INT131 for NASH treatment has been planned.

## 4. Dipeptidyl Peptidase-4 Inhibitors

Dipeptidyl peptidase-4 (DPP-4) inhibitors exert their glucose-lowering effects primarily by blocking the enzyme DPP-4, which is involved in the degradation of incretins, including glucagon-like peptide-1 (GLP-1) and glucose-dependent insulinotropic polypeptide (GIP). Serum DPP-4 levels have been reported to be elevated in NASH patients, as well as correlated with hepatic steatosis and the histopathological grade of NASH. Similarly, circulating DPP-4 concentrations are positively associated with liver fibrosis and hepatocyte apoptosis. Such findings have supported the notion that DPP-4 inhibitors may improve the histological features of NAFLD and NASH. Unfortunately, there is conflicting evidence showing the efficacy of DPP-4 inhibitors in NASH and NAFLD patients with T2D, although the number of patients involved in these studies is relatively small [33]. Evogliptin (DA-1229, Suganon™), a novel DPP-4 inhibitor, was developed by Dong-A ST [34]. However, treatment with saxagliptin, a DPP-4 inhibitor, is associated with an increased risk or hospitalization for heart failure (HF) [35]. Another safety concern is that the use of DPP-4 inhibitor might be associated with an increased risk of cholangiocarcinoma (hazard ratio (HR) 1.77, 95% CI 1.04–3.01) [36] or inflammatory bowel disease (HR 1.75, 95% CI 1.22–2.49) [37] in adults with T2D. Therefore, it is probably best to refrain from administrating DPP-4 inhibitors to T2D patients with NAFLD.

## 5. Glucagon-Like Peptide Receptor Agonists

GLP-1 is a gut-derived incretin hormone that induces weight loss and insulin sensitivity. The blood glucose-lowering action of GLP-1, mediated by its ability to induce insulin secretion and reduce glucagon secretion in a glucose-dependent manner, suppresses appetite and delays gastric emptying. GPL-1 receptor agonists (GLP-1 RAs), which have been used as an antidiabetic agent since 2009, can be an attractive therapeutic option for patients with NASH. GLP-1 RAs have been shown to reduce liver enzymes and oxidative stress and improve liver histology in murine NASH models. The mechanisms of a GLP-1 RA can be explained by improvements in weight and diabetic control. It remains unknown whether GLP-1 RA can directly influence the metabolic phenotype in the liver because conflicting data exist in relation to the presence of GLP-1 receptors on human hepatocytes (Figure 1). GLP-1 RA can act directly on human hepatocytes in vitro, reducing steatosis by decreasing de-novo lipogenesis and increasing fatty acid oxidation [38]. The reported impact of GLP-1 RA on NASH is discussed below.

### 5.1. Liraglutide (Victoza™)

The efficacy of liraglutide, a first-in-class GLP-1RA, has been reported in NASH patients in the West (phase 2 LEAN study [15]) and Japan (LEAN-J study [39]). According to the American Association for the Study of Liver Diseases 2018 practice guidance [40], however, it is premature to consider GLP-1-RA to specifically treat NASH and NAFLD patients without T2D because of insufficient evidence. A phase 3, open-label study is ongoing to compare effects of liraglutide and bariatric surgery on weight loss, liver function, body composition, insulin resistance, endothelial function, and biomarkers of NASH in obese Asian adults (CGH-LiNASH, NCT02654665). In a double-blind trial (LEADER trial) [41], 9340 patients with T2D and high cardiovascular risk were assigned to receive liraglutide or a placebo. The primary composite outcome in the time-to-event analysis was the first occurrence of death from cardiovascular causes, nonfatal myocardial infarction, or nonfatal stroke. The primary outcome occurred in significantly fewer patients in the liraglutide group (608 of 4668 patients (13.0%)) than in the placebo group (694 of 4672 (14.9%), HR, 0.87, 95% CI 0.78–0.97).

### 5.2. Dulaglutide (Trulicity™)

Because most patients, naïve to injection therapy, will hesitate to undergo daily injection therapy, dulaglutide has some advantages, such as weekly injection, disposable and prefilled devices, and safety profiles similar to those of other GLP-1 RAs [42]. Sub-analyses of the AWARD program (AWARD-1, AWARD-5, AWARD-8, and AWARD-9) have proved that dulaglutide has significantly reduced serum transaminase activity and gamma-glutamyl transpeptidase levels compared with a placebo [43]. The REWIND trial proved that dulaglutide had cardioprotective effects in 9,901 T2D patients who had either a previous cardiovascular event or cardiovascular risk factors. In the REWIND trial, the primary composite outcome was the first occurrence of the composite endpoint of non-fatal myocardial infarction, non-fatal stroke, or death from cardiovascular causes. In the dulaglutide group, 594 (12.0%) patients reached the endpoint at an incidence rate of 2.4 per 100 person-years, while 663 (13.4%) patients reached the endpoint at an incidence rate of 2.7 per 100 person-years in the placebo group (HR 0.88, 95% CI 0.79–0.99; *p* = 0·026). All-cause mortality was similar between groups (536 (10.8%) in the dulaglutide group vs. 592 (12.0%) in the placebo group; HR 0.90, 95% CI 0.80–1.01; *p* = 0.067) [44].

### 5.3. Semaglutide (Ozempic™)

Semaglutide, a novel GLP-1 RA, has been recently approved for diabetic patients in the US, EU, Canada, and Japan. To investigate the effect of semaglutide on NASH, a phase 2 randomized double-blind placebo-controlled trial (RDBPCT) comparing the efficacy and safety of three different doses of semaglutide (once-daily subcutaneous injection) versus placebo in 288 participants with NASH (stage 1–3 fibrosis) is ongoing (SEMA-NASH study, NCT02970942). Initial results from the study are expected in May 2020, with the study completion anticipated in July 2020. Semaglutide has three advantages over other GLP-1 RAs. First, the SUSTAIN-6 trial showed that semaglutide had the potential benefit of the prevention of cardiovascular events [45]. In sub-analyses of the SUSTAIN-6 study, semaglutide has been shown to reduce alanine aminotransferase (ALT) and hypersensitive C-reactive protein [46]. Semaglutide has been proved to be superior to dulaglutide for glucose control and weight loss in T2D patients (SUSTAIN 7 trial) [47]. The SUSTAIN-7 trial is a phase 3b, 40-week, efficacy, and safety trial of 0.5 mg semaglutide versus 0.75 mg dulaglutide and 1.0 mg semaglutide versus 1.5 mg dulaglutide, both once-weekly, as an add-on to metformin in 1,201 people with T2D. The SEMA-MR trial is also ongoing. Finally, oral semaglutide under development has shown significant cardiovascular risk reduction. Novo Nordisk initiated its phase 3a program to study the efficacy of 2.4 mg of semaglutide once-weekly for obesity indications. This study program, which comprises four trials, is expected to be completed in 2020. According to a recent meta-analysis of seven trials consisting of ELIXA (lixisenatide), LEADER (liraglutide), SUSTAIN-6 (semaglutide), EXSCEL (exenatide), Harmony outcomes (albiglutide), REWIND (dulaglutide), and PIONEER 6 (oral semaglutide), treatment with GLP-1 RA has beneficial effects on cardiovascular, mortality, and kidney outcomes in patients with T2D [48]. As a result, GLP-1 RAs will be most promising in the treatment of NASH with T2D [33,49].

## 6. Sodium-Glucose Cotransporter (SGLT) Inhibitors

### 6.1. SGLT2 Inhibitors

Sodium-glucose cotransporter 2 (SGLT2) inhibits glucose reabsorption in the proximal tubule, thus leading to glucouria and plasma glucose reduction (Figure 1). Therefore, SGLT2 inhibitors have become promising therapeutic agents in NASH and NAFLD patients. Several pilot studies have found a significant reduction in transaminase activity, body weight, the fatty liver index, and liver histology (steatosis and fibrosis) in NAFLD patients [50,51,52,53,54,55]. Two open randomized controlled trials (RCTs) have been performed in Japan to compare the efficacy of SGLT2 inhibitor with other oral diabetic agents, including pioglitazone and metformin. Hepatic fat content, which has been evaluated by the liver to spleen ratio on computed tomography image, has been significantly decreased in the luseogliflozin group compared with the metformin group [56]. In another report, comparing the efficacy of ipragliflozin versus pioglitazone in NAFLD patients with T2D, serum ALT levels, HbA1c, and fasting plasma glucose have been similarly reduced in the two treatment groups. Nevertheless, significant reductions in body weight and visceral fat area have been observed only in the ipragliflozin group [57]. Not only HbA1c and transaminase activities but also hepatic fat content evaluated by MRI-hepatic fat fraction have been significantly decreased after a 24-week therapy with luseogliflozin. Although hepatic fibrosis markers have been unchanged, serum ferritin levels have been decreased, and serum albumin has been significantly increased after the treatment (LEAD trial) [58]. In the E-LIFT trial, 50 T2D patients with NAFLD who were at least 40 years old were randomly assigned to receive empagliflozin (10 mg/day) plus their standard medical treatment for T2D, such as metformin and/or insulin, or to receive only their standard treatment without empagliflozin (control group) Their liver fat content was measured using magnetic resonance imaging-proton density fat fraction (MRI-PDFF). After 20 weeks of treatment, the liver fat content of patients receiving empagliflozin decreased from an average of 16.2% to 11.3% (*p* < 0.0001), whereas the control group had only a decrease from 16.4% to 15.6% (*p* = 0.057) [59]. A multicenter, RDBPCT, interventional, and an exploratory pilot study in patients with newly diagnosed T2D is ongoing to evaluate the effects of empagliflozin treatment on hepatocellular lipid content, liver energy metabolism, and body composition (NCT02637973). The effect of SGLT2 inhibitors on NAFLD is also investigated compared with other diabetic agents (metformin or sulfonylurea) (NCT02696941 and NCT02649465). The impact of empagliflozin on liver enzymes (ALT and aspartate aminotransferase [AST]) has been analyzed in the EMPA-REG OUTCOME trial [60]. In this trial, patients with T2D and established cardiovascular disease were randomized to receive 10 mg or 25 mg of empagliflozin or a placebo in addition to standard care. Changes from baseline ALT and AST were assessed in all treated patients (*n* = 7,020). The results were a reduction in ALT and AST with empagliflozin versus placebo, with greater reductions in ALT than AST, in a pattern consistent with the reduction of liver fat. This study also demonstrated that reductions in ALT were greatest in the highest tertile of baseline ALT (placebo-adjusted mean difference at week 28: −4.36 U/L (95% CI −5.51, −3.21); *p* < 0.0001) [61]. A phase 3-RDBRCT study is ongoing to evaluate the histological efficacy and safety of dapagliflozin in NASH (NCT03723252). The study of Dapagliflozin Efficacy and Action in NASH (DEAN) study is now recruiting and will enroll 100 participants. This is a phase 3, multicenter, RDBPCT to assess the efficacy and safety of dapagliflozin for improving biopsy-proven NASH and metabolic risk factors. The DAPA-HF trial has been conducted on the standard of care treatment in patients with HF with reduced ejection fraction, including those with and without T2D [62]. Dapagliflozin has met the primary composite endpoint with a statistically-significant and clinically-meaningful reduction of cardiovascular death or the worsening of heart failure (defined as hospitalization or an urgent heart failure visit), compared with a placebo. Remogliflozin-etabonate (KGT-1681), a novel SGLT2 inhibitor, has been shown to reduce liver fat content and transaminase activities in diet-induced obese male mice [63]. Avolynt is developing remogliflozin-etabonate for NASH and initiated remogliflozin etabonate for NASH patients in 2016. Remogliflozin has significantly reduced non-invasive fibrosis markers, such as the fibrosis-4 (FIB-4) index and NAFLD fibrosis score (NFS). However, remogliflozin has been discontinued because of evaluating circumstances, including the development of SGLT2 inhibitors by competitors. Ertugliflozin (MK-8835/PF-04971729, Steglatro™) is an orally active SGLT2 inhibitor being developed by Merck and Pfizer as a treatment for T2D (VERTIS MONO extension study) [64].

### 6.2. SGLT1 Inhibitors (KGA-3235)

Sodium-glucose cotransporter 1 (SGLT1) plays an important role in the intestinal absorption of glucose and, to a smaller extent, the renal reabsorption of glucose (Figure 1). The inhibition of SGLT1 may represent an interesting therapeutic option in patients with diabetes. Kissei has discovered the SGLT1 inhibitor-KGA-3235 for diabetes and licensed the development and marketing rights of the agent in the US and Europe to GlaxoSmithKline. With regard to the development of SGLT inhibitors, GlaxoSmithKline has decided to continue to develop KGA-3235.

### 6.3. Dual SGLT1/2 Inhibitors

Dual SGLT1/2 inhibitors, such as sotagliflozin (LX4211, Lexicon) and licogliflozin (LIK066, Novartis), are now under development. Sotagliflozin has been established to be effective in T1DM patients uncontrolled with insulin [65]. Although phase 2 and 3 trials are ongoing for the treatment of patients with HF and T2D, respectively, NASH studies have never been considered. Licogliflozin is a once-daily, oral, SGLT1/2 dual inhibitor. A phase 2a study in 110 obese patients with NASH stages F1-F3 has been completed (NCT03205150). The primary outcome was the change from baseline in ALT at week 12. Enrolled patients were randomly divided into three groups, including 30 mg/day licogliflozin (*n* = 44), 150 mg/day licogliflozin (*n* = 44), and placebo (*n* = 22). In the Liver Meeting 2019, Harrison and colleagues demonstrated dose-dependent improvement in liver enzymes and PDFF associated with weight loss. However, 76.5% of patients in the higher dose group experienced diarrhea versus ~40% for the placebo and low dose groups.

## 7. Mitochondrial Target of Thiazolidinedione

MSDC-0602K (Cirius Therapeutics) is a next-generation, small-molecule, PPARγ-sparing thiazolidinedione that is the mitochondrial target of thiazolidinedione(mTOT)-modulating insulin sensitizer. It is taken orally and once daily. Pyruvate is produced in the cytosol but must enter the mitochondrial matrix by the mitochondrial pyruvate carrier. MSDC-0602K is designed to preferentially target the carrier while minimizing direct binding to the transcriptional factor PPAR-γ (Figure 1). MSDC-0602K has been shown to be protective in NASH animal models. A phase 2b study to evaluate the safety, tolerability, and efficacy of MSDC 0602K in patients with biopsy-proven NASH (stages F1–F3) has been reported (EMMINENCE trial, NCT02784444). Patients were randomly assigned to a placebo (*n* = 94), or 62.5 mg (*n* = 99), 125 mg (*n* = 98), or 250 mg (*n* = 101) of MSDC-0602K [16]. Initiated in July 2016, the EMMINENCE trial enrolled 392 patients with an average baseline NAS of 5.3. The primary outcome was NAS reduction of 2 points or more with a ≥ 1-point reduction in either ballooning or inflammation without worsening the fibrosis stage. According to the interim results from the EMMINENCE trial, histological improvement in the MSDC-0602K group was not different from the placebo group (Table 2). However, observations showed significant improvement at 6 months in fasting glucose, HbA1c, and insulin levels and HOMA-IR score at the 125-mg and 250-mg dose levels, in addition to a significant reduction in ALT and AST levels [16]. Unfortunately, the overall adverse event (AE) rate was similar across the placebo and all doses of MSDC-0602. In 2020, a phase 3 study (MMONARCh) will be initiated.

## 8. Fibroblast Growth Factor-21 (Pegbelfermin, BMS-986036)

Fibroblast growth factor-21 (FGF-21), a non-mitogenic hormone, is a key regulator of energy metabolism. FGF-21 may play a protective role against NAFLD. In the past decade, FGF-21 has emerged as a metabolic regulator that, under certain stimuli (i.e., fasting, ketogenic diet, and cold exposure) can increase energy expenditure, stimulate insulin sensitivity, and induce weight loss when administered as a pharmacological treatment [66,67] (Figure 1). The plasma FGF-21 level correlates with the severity of NASH, in particular of fibrosis, in patients with NASH [68]. An RCT with a small group of obese T2D patients with FGF-21 has found significant improvement in lipid profiles and insulin resistance, as well as weight loss and increased adiponectin levels [69]. Because endogenous FGF-21 has a short half-life of 1–2 h, it is essential to create long-acting FGF-21 analogs to enable up to weekly dosing. Pegbelfermin is a polyethylene glycol-conjugated recombinant analog of human FGF-21. A multicenter, phase 2a-RDBPCT of pegbelfermin in adults with BMI ≥ 25 kg/m^2^, biopsy-proven NASH (F1–F3), and hepatic fat fraction ≥ 10% (assessed by MRI-PDFF) for 16 weeks has been completed (NCT02413372). Patients received subcutaneous injections of 10 mg pegbelfermin daily (*n* = 25), 20 mg pegbelfermin weekly (*n* = 23), or a placebo daily (*n* = 26) for 16 weeks. The primary endpoint was the absolute change in MRI-PDFF at week 16. At week 16, both dosing regimens of pegbelfermin (10 mg daily or 20 mg weekly) significantly reduced liver fat content versus the placebo (6.8% and 5.2%, versus 1.3%, respectively, *p* = 0.0004 and *p* = 0.008). Both dosing regimens also reduced N-terminal type III collagen propeptide, a novel fibrosis biomarker [70], liver stiffness evaluated by MRE, and transaminase levels. Lipid profiles were also improved in the treatment groups. Overall, pegbelfermin had a favorable safety profile with no serious AEs and no discontinuations due to AEs (NCT02413372) [71]. Unfortunately, 12-week pegbelfermin treatment has not been shown to impact HbA1c concentrations in another randomized phase 2 study [72]. International phase 2b studies (FALCON 1 and 2) of pegbelfermin, for the treatment of NASH stages 3 (NCT03486899) and 4 (NCT03486912), are ongoing.

## 9. Ketohexokinase Inhibitor

In NAFLD patients in NAFLD Clinical Research Network, daily fructose ingestion is associated with reduced hepatic steatosis but increased fibrosis after controlling for age, gender, BMI, and total calorie intake [73]. Fructose rapidly enriches glycolytic metabolite pools, leading to activation of the carbohydrate response element-binding protein, a highly lipogenic transcription factor, which can promote steatosis and insulin resistance. Ketohexokinase (PF-06835919) is the principal enzyme responsible for fructose metabolism. Ketohexokinase catalyzes the conversion of fructose to fructose-1-phosphate, which mediates dietary sugar into the pathway of de novo lipogenesis (Figure 1). Ketohexokinase inhibitor may reduce HbA1c levels and insulin resistance. A phase 2a-RDBPCT is ongoing to evaluate the safety, tolerability, and pharmacodynamics of ketohexokinase inhibitor (PF-06835919) administered once daily for 6 weeks in adults with NAFLD (NCT03256526). In this study, 47 patients completed the course. Mean changes of hepatic fat evaluated by MRI-PDFF in placebo (*n* = 17), 75 mg PF-06835919 (*n* = 17), and 300 mg PF-06835919 (*n* = 13) were −7.97 ± 24.521%, 2.84 ± 22.246%, and −25.43 ± 22.434%, respectively [74]. No AEs were reported in this 6-week trial.

## 10. Novel Antidiabetic Agents

A glucagon receptor (GCGR) agonist is being investigated for the treatment of NAFLD due to its appetite and food intake-reducing effects, as well as its ability to increase lipid oxidation and thermogenesis. MEDI0382 [75], a GLP-1/GCGR dual agonist, dramatically reduces hepatic collagen in a NASH mouse model. Hepatic lipid has been reduced by 40% with MEDI0382 treatment (*p* < 0.0001), which is more effective than liraglutide or a switch to a low-fat diet. Hepatic collagen, quantified by type 1 collagen immunohistochemistry, is increased more than 2-fold with NASH and is reduced by 40% in MEDI0382-treated mice (*p* = 0.005). A phase 2a RDBPCT showed that MEDI0382 had the potential to deliver clinically meaningful reductions in blood glucose and body weight in obese or overweight individuals with T2D [75]. Oxyntomodulin (JNJ-64565111), which binds to both the GLP-1 receptor and the GCGR, improves steatohepatitis and liver regeneration in mice [76]. Several studies of oxyntomodulin (phase 1, Jansen) is ongoing for T2D or obese patients. SAR425899 [77] is a novel dual GLP-1/GCGR agonist. A 52-week, phase 2 RDBPCT to assess the efficacy and safety of SAR425899 for the treatment of NASH was scheduled but withdrawn by the sponsor for reasons unrelated to safety (RESTORE, NCT03437720). Regarding GCGR agonists, AEs, such as hyperglucagonemia and pancreatic alpha cell hypertrophy, are well known.

Tirzepatide (TZP, LY3298176, Lilly), a dual GIP and GLP-1 RA, has shown significantly better efficacy with regard to glucose control and weight loss than dulaglutide, with an acceptable safety and tolerability profile [78]. Results from a sub-analysis have also shown that treatment with TZP leads to larger ALT reduction in the TZP group (10 or 15 mg/day) compared with dulaglutide (1.5 mg/week). The TZP group (10 or 15 mg/day) has shown adiponectin elevation compared with the placebo group (Figure 2). A phase 2b study of TZP for NASH will be planned in 2020.

Imeglimin, the first in a new tetrahydrotriazine-containing class of oral antidiabetic agents, has effects on the liver, muscles, and pancreas—three key organs involved in T2D pathophysiology—through mechanisms suspected to involving the mitochondria and reduced oxidative stress (Figure 1) [79]. Imeglimin improves glucose uptake by impaired muscle tissue, excess hepatic gluconeogenesis, and increased beta-cell apoptosis [80]. Imeglimin has been shown to reduce serum transaminase levels in a sub-analysis of a Japanese phase 2 trial. A phase 3 trial in Japan, trials of Imeglimin for efficacy and safety, will enroll 1100 patients with T2D. Interim analysis has reported a significant reduction in HbA1c. Imeglimin will continue to be studied for the treatment of NASH.

The G-protein-coupled receptor 119 (GPR119, APD778) is a promising target for T2D. Although the role of GPR119 activation in hepatic steatosis and its precise mechanism has not been investigated [81], the GPR119 ligand alleviates hepatic steatosis by inhibiting sterol responsive element binding protein-1-mediated lipogenesis in hepatocytes (Figure 1). Co-administration of GPR119 with linagliptin prevents the progression of NASH in mice models [82,83].

## 11. Combination Therapies

A recent study using network analysis has shown that the use of SGLT2 inhibitors or GLP-1 RA is associated with mortality lower than DPP-4 inhibitors [84]. Therefore, we believe that SGLT2 inhibitors and GLP-1 RA will also become central players in the treatment of T2D patients with NASH. Though the combination of SGLT2/GLP-1RA has already been evaluated in patients with T2D in several studies (AWARD 10 [85], Duration 8 [86], and AGATE [87]), there have been no studies evaluating the efficacy of combination therapy with these agents in the treatment of NASH. The potentially complementary mechanisms of action, and the cardio- and nephroprotective effects demonstrated by molecules of both classes make these drugs potentially useful even as an add-on to each other (Table 3) [88,89,90,91].

Cenicriviroc is an oral inhibitor of the C-C motif chemokine receptor-2/5, which plays an important role in the hepatic recruitment of macrophages [92]. AURORA, a phase 3 study [93], will evaluate the effects of cenicriviroc on hepatic fibrosis in 2000 patients with NASH and is ongoing. A phase 2a, multi-center RDBPCT of cenicriviroc is being conducted with approximately 50 adult obese subjects (BMI ≥ 30 kg/m^2^) with prediabetes or T2D and suspected NAFLD (ORION study, NCT02330549). Other combination therapies are planned, including antidiabetic drug plus metabolic modifiers (PPARδ agonist or farnesoid X receptor agonist) or anti-inflammatory agents, such as cenicriviroc (Figure 3).

## 12. Addressing Comorbid Metabolic Disorders

NAFLD patients with T2D are likely to have a high incidence of comorbid metabolic disorders, such as hypertension, dyslipidemia, hyperuricemia, and cardiometabolic diseases [27]. Clinical trials of statins as a treatment for NASH are limited and have shown inconsistent results, with liver enzymes improving modestly or not at all and variable effects on histology when this was assessed [94,95,96,97]. One small RCT has not demonstrated a benefit of simvastatin, reducing liver enzymes or liver histology [98]. However, statin may reduce the risk of hepatocarcinogenesis in diabetic patients [99,100]. In patients with elevated low density lipoprotein-cholesterol (LDL-C) levels, statin use can be recommended. Ezetimibe has been shown to have no effect on NASH and NAFLD in the MOZART trial [101]. In NAFLD patients with elevated triglyceride, ω3 fatty acid can be recommended without showing any clinical efficacy for NAFLD, as shown in the WELCOME trial [102]. Pemafibrate, a novel selective PPAR α modulator, was approved in Japan in 2017. Pemafibrate, which has been shown to improve liver pathology in a diet-induced rodent model of NASH [103], will become a promising therapeutic agent for human NASH. In Japan, a clinical phase 2 trial for the treatment of NAFLD and NASH with ≥ 10% on MRI-PDFF and ≥ 2.5 kPa on MRE is ongoing. The primary endpoint is percentage change from baseline to week 24 in liver fat content by MRI-PDFF (NCT03350165). Angiotensin receptor blockers, a class of anti-hypertensive drugs, are potential therapeutic agents for NAFLD because of their anti-inflammatory or antifibrotic actions [104]. Current evidence is insufficient to support the efficacy of angiotensin receptor blockers in managing fibrosis in NAFLD patients [105]. Although no approved drugs exist for NAFLD patients with hypertension, a phase 2 study of mineral corticoid receptor antagonist (Aparerenone) is ongoing.

## 13. Precision Medicine in the Treatment Strategy of NASH

Considering the current failures of phase 2 and 3 studies in the drug pipelines for a large population, the perspective of precision medicine may be required in the treatment strategy for NASH. In the middle-aged population, NAFLD is more prevalent in males compared with females. However, NASH with severe fibrosis is frequent in menopausal women. Sex differences may be responsible for treatment efficacy. Asian women are more insulin-sensitive than men at the level of adipose tissue. However, muscle insulin sensitivity is not different between the sexes [106]. Future clinical trials should be designed to test drug efficacy and safety according to sex, age, reproductive stage (i.e., menopause), and synthetic hormone use [107]. We should be prepared to provide any sex-specific therapeutic approaches to NASH patients with T2D. Moreover, genetic and epigenetic factors identifying specific sub-phenotypes of NAFLD can predict the individual response to pharmacological therapies. Several preliminary reports have shown that PNPLA3 (patatin-like phospholipase domain containing 3) polymorphism may influence the efficacy of pharmacotherapies, including antidiabetic treatment [108,109,110]. Studies of gene-targeted therapeutic approaches for NAFLD are in progress [111].

## 14. Clinical Endpoints

Primary endpoints are variable, according to studies as mentioned in this article, including hepatic enzymes (ALT), hepatic fat content (measured by MRI-PDFF), NASH resolution, reduction in NAS (≥2 points), and hepatic fibrosis (determined by histology or MRE) (Table 4). Noninvasive tests, such as the FIB-4 index, NFS, and enhanced liver fibrosis scores, have not been established to evaluate drug efficacy, although a few longitudinal studies have shown the efficacy of the FIB-4 index or NFS as monitoring tools. Antidiabetic drugs, such as SGLT2 inhibitors and GLP-1 RA, have shown their improved overall survival in T2D patient by a variety of global large population studies (EMPA-REG outcome [60], CANVAS program [112], DECLAIR [113], CREDENCE [114], and DAPA-HF [62]). However, no diabetic agents have been proved to improve liver-related mortality. In future trials, over-all mortality or liver-related mortality should be evaluated as final endpoints in NAFLD with T2D.

## 15. Conclusions

To prevent liver-related morbidity and mortality in NASH patients, those with fibrosis should be considered for pharmacotherapies in addition to conventional dietary interventions. Diabetic NASH patients should be preferentially treated with novel drugs licensed for diabetes treatment, such as GLP-1RA and SGLT2 inhibitors because these agents also have cardioprotective and renoprotective efficacy. Currently, several innovative diabetic agents are in the drug pipeline for NASH worldwide, including mTOT, GLP-1/GCGR agonist, GIP/GLP-1 agonist, and imeglimin. Among a variety of SGLT2 inhibitors, dapagliflozin has entered phase 3 trials (DEAN study). SGLT1/2 dual inhibitors (licogliflozin) are also expected. Cost-effectiveness data and patient-centered benefits are also required to position-specific medications in practical guidelines for the treatment of NASH.

## Figures and Tables

**Figure 1 ijms-21-01907-f001:**
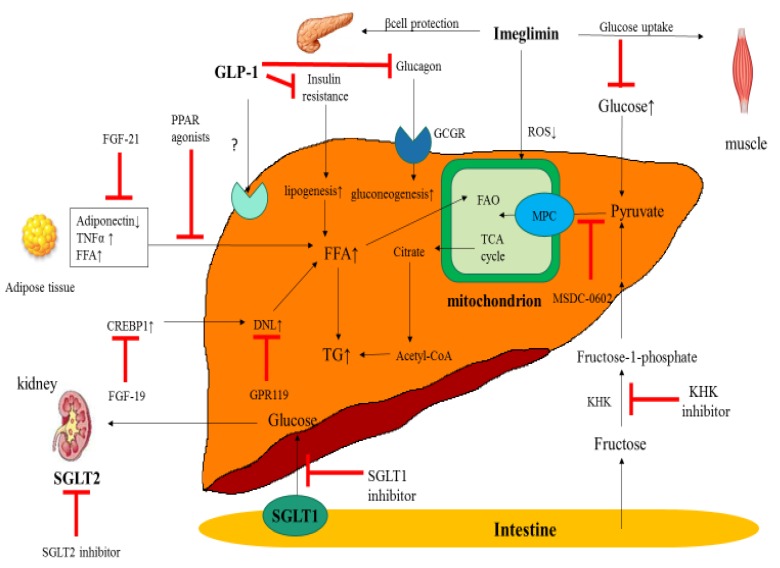
Mechanisms of antidiabetic therapies for nonalcoholic steatohepatitis (NASH). GLP-1: glucagon-like peptide, DNL: de novo lipogenesis, TNFα: tumor necrosis factor α, FFA: free fatty acid, TG: triglyceride, KHK: ketohexokinase, GCGR: glucagon receptor, GPR: G-protein-coupled receptor, MPC: mitochondrial pyruvate carrier, SGLT: sodium-glucose cotransporter, TCA: tricarboxylic acid, ROS: reactive oxygen species, PPAR: peroxisome proliferator-activated receptor, FGF: fibroblast growth factor, CREBP: cAMP-response element-binding protein, FAO: fatty acid oxidation.

**Figure 2 ijms-21-01907-f002:**
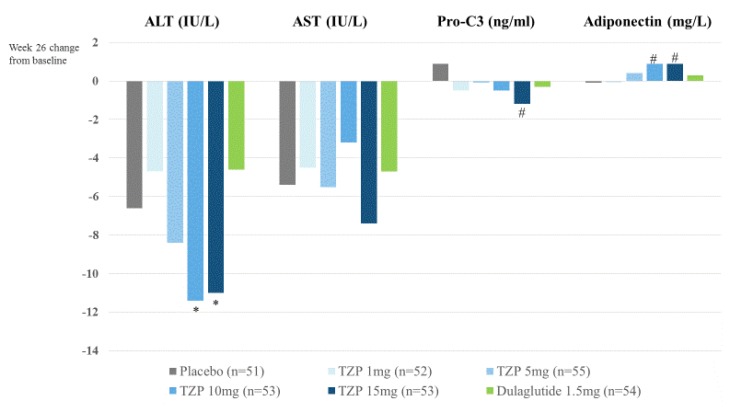
Sub-analyses of efficacy and safety of tirzepatide (TZP) in patients with type 2 diabetes: a randomized, placebo-controlled, and dulaglutide-controlled phase 2 trial [57]. * *p* < 0.05 change from baseline vs. Dulaglutide. # *p* < 0.05 change from baseline vs. placebo.

**Figure 3 ijms-21-01907-f003:**
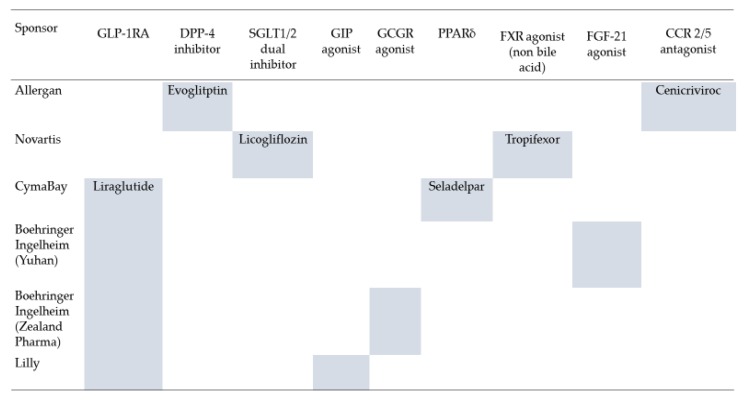
Drug pipelines of NASH “Combo”, including antidiabetic drugs. GLP-1RA: glucagon-like peptide receptor agonist, DPP-4: dipeptidyl peptidase-4, SGLT: sodium-glucose cotransporter, GIP: gastrointestinal peptide, GCGR: glucagon receptor, PPAR: peroxisome proliferator-activated receptor, FXR: farnesoid X receptor, FGF-21: fibroblast growth factor-21, CCR2/5: C-C motif chemokine receptor-2/5.

**Table 1 ijms-21-01907-t001:** Antidiabetic drugs for nonalcoholic steatohepatitis (NASH) under development.

Drug Action	Study Name (NCT)	Drug Name	Phase	Route	Dose (Per Day)	Patients	Therapy Period	N	Primary Outcome	Status
GLP-1RA	LEAN (NCT01237119)	Liraglutide	2	Injection daily	1.8 mg placebo	Obese NASH	24 wk	52	A decrease in NAS of at least 2 points with no worsening fibrosis	Published [15]
	CGH-LiNASH (NCT02654665)		3		0.6 →3.0 mg	Obese NASH	12 mo.	36	Improvement in NASH	Recruiting
	SEMA-NASH (NCT02970942)	Semaglutide	2	Injection daily	0.1 mg 0.2 mg 0.4 mg placebo	NASH stage 1-3	72 wk.	288	NASH resolution without worsening of fibrosis	On going
	D-LIFT (NCT03590626)	Dulaglutide	-	Injection weekly	0.75→1.5 mg per wk.	NAFLD with T2D	24 wk.	60	Change in liver fat quantified by MRI-PDFF	Recruiting
KHK inhibitor	(NCT03969719)	PF-06835919	2a	Oral	150 mg 300 mg	NASH with T2D receiving MTF	16 wk.	150	Percent change from baseline in whole liver fat Change from baseline in HbA1c	Not yet Recruiting
mTOT	EMMINENCE (NCT02784444)	MSDC-0602K	2a	Oral	62.5 mg 125 mg 250 mg Placebo	Obese NASH stage 2/3	12 mo.	380	Reduction in NAS of 2 points or more.	Published [16]
	MMONARCh (NCT03970031)		3		Not shown	NASH with T2D	31 mo.	3600	Mean change in HbA1c from Baseline Histological resolution of NASH	Not yet Recruiting
SGLT 1/2 inhibitor	(NCT03205150)	Licogliflozin	2a	Oral	30 mg 150 mg Placebo	Obese NASH stage 1–3	12 wk.	110	Change from baseline in ALT	Recruiting
SGLT2 inhibitor	DEAN (NCT03723252)	Dapagliflozin	3	Oral	10 mg Placebo	NASH with T2D	12 mo.	100	Scored liver histological improvement	On going

KHK: ketohexokinase, MTF: metformin, GLP-1RA: glucagon-like peptide receptor agonist, mTOT: mitochondrial target of thiazolidinedione modulating, MRI-PDFF: magnetic resonance imaging proton density fat fraction, NAS: nonalcoholic fatty liver disease (NAFLD) activity score, T2D: type 2 diabetes, SGLT: sodium-glucose cotransporter.

**Table 2 ijms-21-01907-t002:** MSDC-0602K effects in NASH liver histology (EMMINENCE trial, Phase 2a).

		Placebo (*n* = 94)	MSDC-0602K			*p* Value
			62.5 mg (*n* = 99)	125 mg (*n* = 98)	250 mg (*n* = 101)	
Primary endpoint	NAS improvement *	29.7%	29.8%	32.9%	39.5%	NS
Secondary endpoints	NASH resolution	20.3%	29.8%	32.9%	39.5%	NS
	Fibrosis improvement	21.6%	23.8%	28.0%	29.1%	NS

NAS: NAFLD activity score, NASH: nonalcoholic steatohepatitis. * NAS improvement was defined by “NAS of 2 points or more with a ≥ 1 point reduction in either ballooning or inflammation without worsening fibrosis stage.”

**Table 3 ijms-21-01907-t003:** The potential benefit of the combination of GLP-1RA/SGLT2 inhibitor therapy [88,89,90,91].

Action	GLP-1RA	SGLT2 Inhibitor	Combination Therapies
Appetite	↓	↑?	↓
Body weight	↓↓	↓	↓↓↓
Insulin secretion	↑	↓	↑?
Glucagon	↓	→	→?
Body pressure	↓	↓	↓↓
Bone mineral density	↑?	↓?	→?
Muscle volume	↑?	↓?	→?
Amputation risk	→	↑ or →?	→?
Heart failure	↓	↓	↓↓
Renoprotection	+	+	++
Hepatic fat quantity	↓	↓	↓↓
Hepatic fibrosis	↓	↓?	↓?

GLP-1RA: glucagon-like peptide-1 receptor agonist, SGLT2: sodium-glucose cotransporter. ↑: increase, ↓: decrease, →: no change, ?: uncertain, +: positive effect.

**Table 4 ijms-21-01907-t004:** A variety of endpoints of antidiabetic agents for NASH according to study design (phase 2/3).

Endpoints	Parameters		Phase
Liver histology	• Steatosis (0–3)		Phase 2b/3
	• Inflammation (0–3)		
	• Ballooning (0–2)		
	• NAS > 2 points reduction without worsening fibrosis		
	• NASH resolution without worsening fibrosis		
	• Fibrosis improvement > 1 stage without worsening NASH		
Imaging studies	VCTE	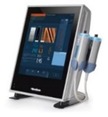 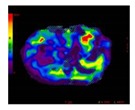	Phase 2a
	CAP: steatosis		
	LSM: stiffness		
	MRI		
	PDFF: steatosis ≥ 30% relative reduction		
	MRE: stiffness ≥ 15% reduction		
	Multiparametric MRI: inflammation, ballooning		
Biochemistry and scoring systems	• AST, ALT		Pilot/Phase 2a
	• HbA1c		
	• FIB-4 index		
	• NFS		
	• APRI		
	• ELF test		
	• Pro-C3		

NAS: NAFLD activity score, NASH: nonalcoholic steatohepatitis, VCTE: vibration-controlled transient elastography, CAP: controlled attenuation parameter, LSM: liver stiffness measurement, MRI: magnetic resonance imaging, PDFF: proton density fat fraction, MRE: magnetic resonance elastography, AST: aspartate aminotransferase, ALT: alanine aminotransferase, HbA1c: glycated hemoglobin, FIB-4: fibrosis-4, NFS: NAFLD fibrosis score, APRI: AST to platelet ratio index, ELF: enhanced liver fibrosis, Pro-C3: true collagen type III formation.

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
