# Peer review of "Antidiabetic Therapy in the Treatment of Nonalcoholic Steatohepatitis"

_ijms, 2020, doi:10.3390/ijms21061907_

Round 1

Reviewer 1 Report

Comments

The authors reviewed about diabetic therapy in the treatment of NASH. It covers current anti-DM drugs (pioglitazone, GLP-1RA, and SGLT2i) as well as upcoming drugs such as Fgf21, and ketohexokinase inhibitors, glucagon receptor agonists, etc. This topic matches the scope of IJMS.

The treatment of NASH is one of most important topic in hepatology, I think that this review will be beneficial for many readers of IJMS. However, I have some concerns.

1. They should describe the summary of markers estimating the improvement of NASH and NAFLD.

2. They should represent one shema about the mechanism how each drugs affect the pathogenesis of NASH.

3. They should describe the molecular mechanism of each drugs briefly.

I think that readers of IJMS are not always physician that see diabetic patients. Basic scientists will also read this review. They help many readers to understand their effect on NASH.

Based on the mechanism, following questions are postulated.

1) There are no GLP-1 Receptor in liver. Do the authors think the mechanism how GLP-1RA affect NASH?

2) do the authors think the mechanism how SGLT2i/GLP-1RA combination affect NASH?

3) Liver lacks a receptor for FGF21 and FGF21 has no direct effect in liver.

Do they think the mechanism how FGF21 reduce liver fat?

4) Many readers will not know ketohexokinase, mitochondrial target of thiazolidinedione, and imeglimin. They should briefly describe about these drugs (effectiveness, mechanism of action, side effect?).

Regarding ketohexokinase, they at first should describe the relationship between excess fructose consumption and obesity related diseases such as NAFLD. It is better to describe about side effect that are seen in ketohexokinase deficiency, too.

Regarding mitochondrial target of thiazolidinedione and other new drugs, too, shema presentation about its mechanism will help readers understand the mechanism of action.

4. They did not cite paper about the no effect of metformin on NAFLD. Please cite reference.

Line 79 ; but it has no effect on NAFLD.

5. Table 1 is strange. Name of drugs (GLP-1RA, SGLT2i, etc) is not a “mechanism”. They shoud change this into appropriate name.

6. Reference 50 is not correct. This is a webpage about this preprint.

Please correct it.

7. About Section2.

How about SPARMalpha (pemafibrate) or PPARa agonist? Although they are not anti-DM drugs, they have some benefit for NASH.

About Section 4

Liraglutide has excellent data (leader trial) about cardioprotective and renoprotective effects, but they did not describe them. In contrast, they described REWIND trial by dulaglutide. They should fairly describe about Leader trial, too. I think that this section undermine confidence in this review.

About Section 10

Regarding glucagon receptor agonist, side effects such as hyperglucagonemia and pancreatic alpha cell hypertropy are well known. They should describe side effects, too.

About Section 6 and 11

Section 6 (SGLT2/GLP1RA combination) and section 11 (combination therapy) should be combined.

Author Response

Reviewer 1

  1. They should describe the summary of markers estimating the improvement of NASH and NAFLD.

Answers: We summarized markers estimating the improvement of NASH and NAFLD (Table 4.). We discussed this point at section 14. “Clinical endpoints (revised manuscript, Figure 1).

  1. They should represent one schema about the mechanism how each drugs affect the pathogenesis of NASH.

Answers: We represented one schema showing mechanism how each drugs affect the pathogenesis of NASH (revised manuscript, Figure 1).

  1. They should describe the molecular mechanism of each drugs briefly. I think that readers of IJMS are not always physician that see diabetic patients. Basic scientists will also read this review. They help many readers to understand their effect on NASH. Based on the mechanism, following questions are postulated.

Answers: As you suggested, we described the molecular mechanism of each drugs briefly throughout this manuscript.

  1. There are no GLP-1 Receptor in liver. Do the authors think the mechanism how GLP-1RA affect NASH?

Answer: Though precise mechanisms remains unknown, plausible explanations were shown (revised manuscript, line 138-149).

  1. do the authors think the mechanism how SGLT2i/GLP-1RA combination affect NASH?

Answer: The combination therapy of SGLT2i/GLP-1RA is the most promising therapy in NAFLD with T2DM or severe obesity. Plausible mechanisms of these combination therapies are shown in Table 3 (revised manuscript).

3) Liver lacks a receptor for FGF21 and FGF21 has no direct effect in liver.

Do they think the mechanism how FGF21 reduce liver fat?

Answer: Plausible mechanisms are mentioned in revised manuscript (revised manuscript, line 308-315).

4) Many readers will not know ketohexokinase, mitochondrial target of thiazolidinedione, and imeglimin. They should briefly describe about these drugs (effectiveness, mechanism of action, side effect?).

Regarding ketohexokinase, they at first should describe the relationship between excess fructose consumption and obesity related diseases such as NAFLD. It is better to describe about side effect that are seen in ketohexokinase deficiency, too.

Regarding mitochondrial target of thiazolidinedione and other new drugs, too, shema presentation about its mechanism will help readers understand the mechanism of action.

Answer: We at first described the relationship between excess fructose consumption and NAFLD (revised manuscript, line 333-337). The roles of ketohexokinase (KHK) in NASH pathogenesis were discussed. In a short term trial of KHK, no adverse effects were reported (revised manuscript, line 345). The action of mitochondrial target of thiazolidinedione (mTOT) (revised manuscript, line 283-288) and imeglimin were also mentioned (revised manuscript, line 376-380). These mechanisms were summarized in Figure 1.

  1. They did not cite paper about the no effect of metformin on NAFLD. Please cite reference.Line 79 ; but it has no effect on NAFLD.

Answer: As you suggested, we cited four papers as references (revised manuscript, reference No.10-13).

  1. Table 1 is strange. Name of drugs (GLP-1RA, SGLT2i, etc) is not a “mechanism”. They should change this into appropriate name.

Answer: In agreement with you, we revised “drug action” in revised table 1.

  1. Reference 50 is not correct. This is a webpage about this preprint.

Please correct it.

Answer: Thank you for your kind assistance. We corrected it as reference No.78 (revised manuscript).

  1. About Section2.

How about SPARMalpha (pemafibrate) or PPARa agonist? Although they are not anti-DM drugs, they have some benefit for NASH.

Answer: As you suggested, pemafibrate is the promising agent, although it is not antidiabetic drugs. We mentioned a phase 2 trial of pemafibrate is now ongoing in Japan for NAFLD in the section 12 (revised manuscript, line 431-435).

About Section 4

Liraglutide has excellent data (leader trial) about cardioprotective and renoprotective effects, but they did not describe them. In contrast, they described REWIND trial by dulaglutide. They should fairly describe about Leader trial, too. I think that this section undermine confidence in this review.

Answer: In agreement with you, we discussed regarding LEADER trial (revised manuscript, line 166-170).

About Section 10

Regarding glucagon receptor agonist, side effects such as hyperglucagonemia and pancreatic alpha cell hypertrophy are well known. They should describe side effects, too.

Answer: In agreement with you, we added adverse effects of glucagon receptor agonist, side effects such as hyperglucagonemia and pancreatic alpha cell hypertrophy (revised manuscript, line 360-361).

About Section 6 and 11

Section 6 (SGLT2/GLP1RA combination) and section 11 (combination therapy) should be combined.

Answer: In agreement with you, we combined Section 6 and Section 11.

Reviewer 2 Report

GENERAL COMMENT
This study, which addresses a highly relevant clinical topic, is rich of novel information and, therefore, deserves full consideration. Conflicting with the high academic standard of the eminent authors who have elaborated it, however, the style (and, to a limited extent) the contents of the study are amenable to both substantial and formal improvement.

SPECIFIC COMMENT

English is poor and must accurately be edited. Revision by a mothertongue is strongly encouraged. There are innumerable examples of wrong/unclear sentences. Colloquial and informal, long unreferenced statements must also totally be reworked, e.g. "evidences" "The incidence of nonalcoholic steatohepatitis (NASH) is a more severe form of NAFLD." "The incidence of nonalcoholic steatohepatitis (NASH) is a more severe form of NAFLD. NASH refers to liver inflammation due to fat deposition in the liver, has risen dramatically over the last two decades because of growing prevalence of obesity, metabolic syndrome, and type 2 diabetes (T2DM). Also called a "silent killer”, since the symptoms are not manifested in early stages, in some patients, NASH can also progress to fibrosis and cirrhosis over the years, with a high risk for liver failure and hepatocellular carcinoma (HCC). In early stages of NASH, patients generally feel well. However once the disease is more advanced or cirrhosis develops, they begin to experience symptoms such as fatigue, weight loss, and weakness. A person with cirrhosis experiences fluid retention, muscle wasting, bleeding from the intestines, and liver failure. " "NASH can be called “diabetic liver disease (DLD)”. It is estimated that the prevalence of diagnosed NASH will reach 45 billion US dollars by 2027 in US, Japan, and EU 5 (England, France, Germany, Italy, and Spain). Lifestyle interventions such as dietary caloric restriction and exercise are currently the cornerstone of therapy for NASH, can be difficult to achieve and maintain, underscoring the dire need for pharmacotherapy.

The manuscript may also benefit from partial reworking through the inclusion of a quite short section devoted to those aspects of the epidemiology, natural history and the physiopathology of the association of NAFLD with T2D (with reference to the role of nuclear receptors), which are specifically relevant to our understanding of the combination of NAFLD with T2D and to better illustrating the role of the individual classes of drugs. In doing so, there are certain novel data and reviews which may be discussed (Diabetes Care. 2019 Oct 28. pii: dc191113. doi: 10.2337/dc19-1113. [Epub ahead of print]; J Hepatol. 2019 Oct;71(4):793-801.Acta Diabetol. 2019 Apr;56(4):385-396. Medicine (Baltimore). 2017 Sep;96(39):e8179.) In addition, I would sugggest analyzing the following articles Diabetes Metab. 2020 Jan 7. pii: S1262-3636(20)30002-1. doi: 10.1016/j.diabet.2019.12.007.J Hepatol. 2018 Feb;68(2):335-352.J Gastroenterol Hepatol. 2016 May;31(5):936-44. J Clin Endocrinol Metab. 2019 Aug 1;104(8):3327-3336. Diabetes. 2019 Aug;68(8):1681-1691. Diabetes. 2018 Dec;67(12):2485-2493. Medicine (Baltimore). 2018 Sep;97(37):e12356.).

In addition, preliminarily, these Authors should declare which specific primary endpoints - based on our perception of the interaction of NAFLD with T2D as well as of the natural history of disease – should, in general, be prioritized in RCTs on NASH associated with T2D\\\\. For example: decreased serum concentrations of liver enzymes ? resolution of NASH ? Reduction of fibrosis ? Improvement of steatosis ? Decreasing inflammatory grading ? And how should these items be assessed e.g. either invasively or non-invasively. This preliminary discussion may provide a clue for constructing a hierarchy in reporting the various trials which are discussed. This would result in a better development of the part on critical comments (as opposed to simple summary of data) to published studies.

A recent line of research has advocated the use of personalized medicine in NAFLD . In particular, sex differences in NAFLD and related disorders have recently been addressed (J Clin Endocrinol Metab. 2019 Jan 1;104(1):127-136. Hepatology. 2019 Oct;70(4):1457-1469).These Authors must discuss whether, based on data, we are ready to provide any sex-specific therapeutic approaches to patients with T2D.

It is clear that patients with T2D often have other features of the Metabolic Syndrome. A short section must, therefore, address how comorbid metablic disorders (e.g. arterial hypertension and dyslipidemia) should be managed in those NAFLD patients who have T2D.

Author Response

SPECIFIC COMMENT

English is poor and must accurately be edited. Revision by a mother tongue is strongly encouraged. There are innumerable examples of wrong/unclear sentences. Colloquial and informal, long unreferenced statements must also totally be reworked, e.g. "evidences" "The incidence of nonalcoholic steatohepatitis (NASH) is a more severe form of NAFLD." "The incidence of nonalcoholic steatohepatitis (NASH) is a more severe form of NAFLD. NASH refers to liver inflammation due to fat deposition in the liver, has risen dramatically over the last two decades because of growing prevalence of obesity, metabolic syndrome, and type 2 diabetes (T2DM). Also called a "silent killer”, since the symptoms are not manifested in early stages, in some patients, NASH can also progress to fibrosis and cirrhosis over the years, with a high risk for liver failure and hepatocellular carcinoma (HCC). In early stages of NASH, patients generally feel well. However once the disease is more advanced or cirrhosis develops, they begin to experience symptoms such as fatigue, weight loss, and weakness. A person with cirrhosis experiences fluid retention, muscle wasting, bleeding from the intestines, and liver failure. " "NASH can be called “diabetic liver disease (DLD)”. It is estimated that the prevalence of diagnosed NASH will reach 45 billion US dollars by 2027 in US, Japan, and EU 5 (England, France, Germany, Italy, and Spain). Lifestyle interventions such as dietary caloric restriction and exercise are currently the cornerstone of therapy for NASH, can be difficult to achieve and maintain, underscoring the dire need for pharmacotherapy.

Answer: We ordered English editing before submitting revised version.

The manuscript may also benefit from partial reworking through the inclusion of a quite short section devoted to those aspects of the epidemiology, natural history and the physiopathology of the association of NAFLD with T2D (with reference to the role of nuclear receptors), which are specifically relevant to our understanding of the combination of NAFLD with T2D and to better illustrating the role of the individual classes of drugs. In doing so, there are certain novel data and reviews which may be discussed (Diabetes Care. 2019 Oct 28. pii: dc191113. doi: 10.2337/dc19-1113. [Epub ahead of print]; J Hepatol. 2019 Oct;71(4):793-801.Acta Diabetol. 2019 Apr;56(4):385-396. Medicine (Baltimore). 2017 Sep;96(39):e8179.) In addition, I would sugggest analyzing the following articles Diabetes Metab. 2020 Jan 7. pii: S1262-3636(20)30002-1. doi: 10.1016/j.diabet.2019.12.007.J Hepatol. 2018 Feb;68(2):335-352.J Gastroenterol Hepatol. 2016 May;31(5):936-44. J Clin Endocrinol Metab. 2019 Aug 1;104(8):3327-3336. Diabetes. 2019 Aug;68(8):1681-1691. Diabetes. 2018 Dec;67(12):2485-2493. Medicine (Baltimore). 2018 Sep;97(37):e12356.).

Answer: Thank you for your kind assistance. As you suggested, we added the section 2 “Association of T2D with NASH and NAFLD” . We cited all papers you suggested.

In addition, preliminarily, these Authors should declare which specific primary endpoints - based on our perception of the interaction of NAFLD with T2D as well as of the natural history of disease – should, in general, be prioritized in RCTs on NASH associated with T2D\\\\. For example: decreased serum concentrations of liver enzymes ? resolution of NASH ? Reduction of fibrosis ? Improvement of steatosis ? Decreasing inflammatory grading ? And how should these items be assessed e.g. either invasively or non-invasively. This preliminary discussion may provide a clue for constructing a hierarchy in reporting the various trials which are discussed. This would result in a better development of the part on critical comments (as opposed to simple summary of data) to published studies.

Answer: These problems you pointed are important issues in the treatment of NASH. Primary outcomes are variable according to study protocol. We mentioned in the section 14 (revised manuscript, line 423-434, Table 4)

A recent line of research has advocated the use of personalized medicine in NAFLD . In particular, sex differences in NAFLD and related disorders have recently been addressed (J Clin Endocrinol Metab. 2019 Jan 1;104(1):127-136. Hepatology. 2019 Oct;70(4):1457-1469).These Authors must discuss whether, based on data, we are ready to provide any sex-specific therapeutic approaches to patients with T2D.

Answer: In agreement with you, we added section 13 “ Precision medicine in the treatment strategy of NASH” (revised manuscript, line 443-456)

It is clear that patients with T2D often have other features of the Metabolic Syndrome. A short section must, therefore, address how comorbid metabolic disorders (e.g. arterial hypertension and dyslipidemia) should be managed in those NAFLD patients who have T2D.

Answer: In agreement with you, we added section 12 “Addressing comorbid metabolic disorders” (revised manuscript, line 420-441)

Round 2

Reviewer 1 Report

The manuscript was improved and I think that it is beneficial for readers in IJMS. But minor change will be needed. 

  1. ketohexokinase is not a drug action. please correct it as ketohexokinase inhibitor.
  2. line 140. GLP-1RA and GLP-1RAs are confused. please correct GLP-1RA as GLP-1RAs.
  3. The gene symbol of glucagon receptor is GCGR.
  4. Please show references used in Table 3.
  5. In Figure3, the letter is difficult to see. Please change the colors of letter.

Author Response

Thank you for your kind review.

  1. ketohexokinase is not a drug action. please correct it as ketohexokinase inhibitor.

Answer: As you suggested, we revised (revised manuscript, line 338 and 340)

  1. line 140. GLP-1RA and GLP-1RAs are confused. please correct GLP-1RA as GLP-1RAs.

Answer: as you suggested, we revised (revised manuscript, line 140)

  1. The gene symbol of glucagon receptor is GCGR.

Answer: As you suggested, we revised (revised manuscript, line 346)

  1. Please show references used in Table 3.

Answer: As you suggested, we added references [92-95] used in Table 3.

  1. In Figure3, the letter is difficult to see. Please change the colors of letter

Answer: As you suggested, we changed the colors of letter in Figure 3.

Reviewer 2 Report

I would like to thank these authors on accepting my suggestions. Their manuscript is improved.

Author Response

Thank you for your kind review